# Promoting Feature Awareness by Leveraging Collaborators' Usage Habits in Collaborative Editors

Emmanouil Giannisakis*
University of British Columbia
Vancouver, Canada

Jessalyn Alvina†
Université Paris-Saclay, CNRS, Inria
Orsay, France

Andrea Bunt‡
University of Manitoba
Winnipeg, Canada

Parmit Chilana§
Simon Fraser University
Burnaby, Canada

Joanna McGrenere¶
University of British Columbia
Vancouver, Canada

## ABSTRACT

Users often rely on their collaborators to find relevant application features by observing them "over the shoulder" (OTS), usually in a synchronous co-located setting. However, as remote work settings have become more common, users can no longer rely on such in-person interaction with collaborators. Therefore, we investigate designs that help the user become aware of relevant features based on collaborators' feature usage habits. We created five design concepts as video prototypes which varied in five design dimensions: number of active collaborators, number of shared documents, specificity of comparison, user involvement, and goal of the feature awareness. Interviews (N=18) probing the design concepts indicate that collaborator-based feature awareness would be valuable for discovering novel features and producing a consistent style across the shared document, but some users may feel micromanaged or self-conscious. We conclude by reflecting on and expanding our design space and discussing future design directions supporting remote OTS learning.

**Index Terms:** User Interfaces [User Interfaces]: Graphical user interfaces (GUI)—Empirical studies in interaction design;

## 1 INTRODUCTION

Modern software applications offer a large set of features which often include hundreds or thousands of different commands and keyboard shortcuts [35]. As a result, it is challenging for users to be aware of the available features and to identify which ones are relevant to their tasks [25, 60, 66]. Although various support tools and mechanisms exist that aim to raise a user's awareness of features, such as online documentation, tutorials, and videos [41], it has been shown that users tend to prefer *social solutions*, where a user learns about a new feature from other users [20, 40, 71]. Such solutions can draw on different "levels" of social communities, from the global level, often referred to as "the crowd", which includes Q&A forums, all the way down to a more local level, such as an individual in the same institution. For example, users commonly rely on their colleagues to discover relevant features by observing them "over-the-shoulder" (OTS) [60, 70] or by directly asking them for help [40]. This type of serendipitous feature discovery thrives in a synchronous co-located setting as users can leverage their shared work context and users tend to trust their colleagues more than other sources [60].

With the increase in remote work over the past few years [68], especially during the COVID-19 pandemic [31], in-person serendip-

*e-mail: em.giannisakis@gmail.com
†e-mail: jessalyn.alvina@lisn.upsaclay.fr
‡e-mail: bunt@cs.umanitoba.ca
§e-mail: pchilana@cs.sfu.ca
¶e-mail: joanna@cs.ubc.ca

itous interactions are far less frequent today, leaving fewer opportunities for feature discovery among colleagues. Screen sharing could potentially enable synchronous OTS interactions, however, a lack of support for communicating about the interactions makes discovering new features in this setting challenging [60, 72]. Prior work has also proposed tools as solutions that facilitate short synchronous help exchanges [7, 38], or provide additional persistent, asynchronous content [24, 72] (e.g., workflows from individuals). Such tools are useful, but they typically require the user to leave their current application and switch to another one, which can be disruptive for both the learner and expert [60]. Therefore, we wondered how could a user observe and leverage a colleague's software knowledge when working in remote asynchronous situations without having to switch from one application to another?

Our overarching goal is to design in-application tools and techniques that promote feature awareness based on a colleague's software knowledge. We focus on leveraging the user's *direct collaborators* within the context of common document(s) in *collaborative editor applications* (e.g., all the users working on a Google Sheet document) to provide feature awareness from trusted sources, who are working on the same tasks. The popularity of collaborative editors has increased over the past decade as they offer a shared environment for users to work remotely, synchronously, or asynchronously [13, 61].

While there is much design inspiration from other feature awareness solutions in the literature, designs that will satisfy our particular goals are not immediately obvious. For example, some existing solutions recommend features based on system-determined "similar users" across all those who use a given application [58, 59]. These tools provide numerical command usage comparisons [59], which might be acceptable with "crowd-level" comparisons, but users might be less comfortable when comparisons are to known colleagues. Users might be comfortable sharing knowledge with their colleagues through Q&A approaches (e.g., AnswerGarden [1]), but are missing application context. Hence, as a starting point we asked: What are the potential benefits, drawbacks, and design consideration for tools that aim to raise feature awareness by leveraging collaborator usage patterns and shared application documents?

To answer our question, we followed a *Research through Design* [77] approach. This approach focuses on the generation of design artifacts that are used as exemplars to probe people's reactions, attitudes, and perceptions, to produce research findings [77]. We first defined a design space based on the existing literature, our own experiences working in collaborative teams, and a small informal formative study. Our initial design space includes five dimensions (Fig. 1) that range from the number of active collaborators, to the degree of user involvement required by the user. We then generated five different design concepts which intentionally emphasized different aspects of the five design dimensions, and we created corresponding video prototypes [76]. We conducted a semi-structured interview study (N=18) to elicit feedback on the potential benefits and drawbacks of the design concepts and to understand

users perception of points in the design space. We used this feedback to reflect on and expand the design space (Fig. 3).

This paper makes the following contributions: First, we outline five design dimensions to characterize the design space around raising feature awareness based on the user's collaborators in a shared application with common documents. These can be used as a generative resource for creating new tools. Second, we offer five alternative design concepts generated using the design space that showcase how the user's collaborators use the application. Our elicitation study probed and explored the space, identifying where the most promising design opportunities lie as well as limitations of our overall approach to raising feature awareness. For example, participants felt such tools would be valuable for not only discovering novel features but also for identifying features that could help a group of collaborators produce a consistent style across the shared document. That said, they might feel micromanaged and self-conscious. Third, we present concrete design implications and important future considerations for raising feature awareness based on the user's collaborators.

## 2 RELATED WORK

Feature awareness is an important part of software learnability and usability [25]. In this section, we focus on reviewing design efforts around raising feature awareness through social solutions that draw on *user communities* and *individual users*. We also briefly touch on *technical solutions*.

### 2.1 Feature Awareness Based on User Communities

Some prior work in feature awareness has utilized the usage habits of broad user communities such as all users of an application (crowd). CommunityCommands [59] recommends commands by implicitly comparing similar users from the crowd using collaborative filtering algorithms [30, 65]. Patina [58] also utilizes similar users from the crowd to highlight commands within the interface that the user most frequently uses and that other similar users most frequently use. As such, Patina provides a visual feature usage *comparison*. Owl [53, 54] is also a feature recommendation system that compares the usage habits of the users within the same organization as the main user to recommend relevant features. These tools operate on the command level and offer a lightweight way to help users become aware of relevant features. Although these solutions can provide useful feature recommendations while *minimizing the user's involvement*, it can be difficult for the users to assess the usefulness of the highlighted features (i.e., relevancy) as they may not have enough information about the users that the system is based on (i.e., trust on the sources) [60, 75].

Prior work has also focused on recommending workflows (i.e., sequences of commands) based on the community. CADament aims to help users observe other users by providing a viewport to their screens [49], Coscripter [47] allows users to create and share scripts to automate processes within the same enterprise. Other tools [44,74] recommend relevant workflow videos generated from the crowd. These tools can increase the user's understanding of the software's capabilities, but they require the user to stop their current task to see the generated videos. Prior work has also leveraged broader user communities to help the user understand how to use their software. For example, AnswerGarden [1] offers a Q&A repository within the organization while other tools like [8, 16, 29, 57] leverage the knowledge of the broader user community by using widgets that are integrated in the user's applications. For example, in LemonAid [8], the user can select an application widget to see community questions and answers related to that widget. Tools like AnswerGarden can help users get help from their direct collaborators. However, they need to interrupt their current task [60], and also it can be difficult to locate useful answers from past discussions [8].

Our work builds on community-based feature awareness tools that offer lightweight and in-application solutions (e.g., Community-

Commands [59] and Patina [58]). However, instead of focusing on large user communities such as the user's organization or all users of an application, we focus on the close-knit group of a user's collaborators on a shared document. We hypothesize that by focusing on this group, we can avoid challenges that systems based on the broader user community often face, such as understanding the user's goals [2, 9] and finding similar users within the community.

### 2.2 Feature Awareness Based on Individual Users

Some tools aim to mediate the social interaction between two users to help one or both discover relevant features. Users prefer this type of social solutions [40, 71] where, for example, they can get task-specific advice by observing what one of their colleagues are doing "over the shoulder" [69]. Such interactions can be very effective, yet they do not happen frequently [60] because they can be time consuming as well as difficult to coordinate and record [72].

Prior work has aimed to address the issues of coordination. Some systems have focused on helping users to find experts who can respond to their questions [32,36,38] which can minimize the response time [63]. MicroMentor [38] for example, helps the user arrange 3-minute sessions with an expert user. MarmalAid [7] anchors real-time chat conversations to individual graphical widgets of a 3D modeling tool. These tools requires *high involvement* from users, as they have to interrupt their current task to join a video call for the learning exchanges. Other tools aim to help the user find relevant workflows by seeing their colleagues asynchronously. For example, Customizer [72] allows users to see how their colleagues have customized their tools and thus help them find relevant workflows. Some other tools [24, 26, 45] record and extract video that shows the workflow that individual users follow to complete a task. Finally, some tools [22, 28] aim to optimize the synchronous one-to-one interaction, especially in the case of IDEs while users are in pair programming sessions [5]. The main goal of these tools [22, 28] is to help the user understand their collaborators' actions, specifically focusing on their collaborators' changes in the shared document.

The above tools can be effective but also time-consuming and require users to stop their current tasks to interact with other users. Therefore, these tools may be more appropriate for helping users solve more complex issues that go beyond feature awareness. Our work focuses on feature awareness and explores design solutions that aim to minimize user involvement and thus task interruption while taking advantage of the user's direct collaborators.

### 2.3 Technical Solutions to raising feature awareness

Prior work has also proposed technical solutions to raising feature awareness. For example, *tip-of-the-day* tools [19] proactively introduce available functionalities, and *quick assist* [19] (often available IDEs) proposes quick fixes when developers face a problem. These tools propose features that are not necessarily relevant to the user or novel [17]. Other tools highlight features based on the user's current context [11, 12, 18], current actions [33, 37] or command usage history [3, 9, 34]. The challenge with these tools is that their domain knowledge is often predesigned and self-contained without considering community knowledge, which constantly evolves [51]. An exception is QFRecs [39] which bases its recommendations on an application's online documentation, which can be up to date with the newest features. Finally, some tools highlight shortcut alternatives using notifications [23, 64], by integrating shortcut cues within the UI [21, 55], or by using external widgets [42, 48, 56]. While these tools offer reactive, contextual help, prior studies indicate that users tend to learn only a small subset of the available shortcuts. [43, 54].

Our work explores a solution that focuses on collaborators' software usage habits to help the user identify the commands and the keyboard shortcuts that they need to complete their current tasks.

## 3 DESIGN SPACE

### 3.1 Methodology Overview and Rationale

Our review of prior work indicates that raising feature awareness based on the user's collaborators while requiring only modest user involvement is an under-explored space. While there are opportunities to apply design insights from related work on crowd-based approaches or solutions that are based on individual users, how to translate these insights and leverage the unique design opportunities afforded by this new context is unclear. Therefore, to systematically explore this design space, we used a *Research through design (RtD)* [77], an approach in interaction design research that intersects theories and technical opportunities to generate a concrete problem framing and a series of design artifacts (e.g., concepts, prototypes, and documentation of the design process). Prior work on raising feature awareness has often focused on proposing, implementing, and evaluating a single system, with the aim of understanding in depth how the proposed system can benefit the user. In contrast, our approach probes on the potential roles, forms, and values of emerging near-future technology by using more than one design vision, as proposed in other works [14, 62]). Prior work has used a similar approach to investigate the design space around supporting cross-device learnability [4], data legacy [27], and personal data curation [73]. We aim to understand user reactions towards this under-explored problem space, to define concrete design goals, and to generate design implications for future implemented systems.

Our application of the *RtD* approach was as follows: We first carefully generated a set of design dimensions as similarly done in [4, 73]. We generated this set by clustering and mapping insights from prior work, reflecting on the authors' personal experiences, and using findings from an informal formative study. During a series of our research group meetings, we refined these insights into a set of five relevant design dimensions. These dimensions are not meant to be exhaustive, but rather are those that seem to be most prominent based on our review of our insights from prior work and the informal formative study. We then use this set of design dimensions as a generative tool to create five design concepts in the form of video prototypes. Finally we use these design concepts in an interview study to elicit participants reactions towards the problem space and aspects of our design space.

In the remainder of this section, we describe our informal formative study and detail our proposed design space.

### 3.2 Informal Formative Study: Method and Analysis

We conducted an informal formative study with two goals in mind: 1) to understand how users currently learn from each other when collaborating remotely, and 2) to gather initial thoughts on how raising feature awareness based on their collaborators might impact their current practices. We advertised our study on a university mailing list. We recruited 11 participants (6 women and 5 men, 21-30 years old) with diverse occupations (e.g., accountants, data analyst, event planner, etc.), all of whom reported collaborating with others at least once per week using editors like Google Docs.

During a 60-minute Zoom session with each participant, we introduced an interactive prototype[1] that shows feature recommendations within an editor that differ in terms of 1) the user community from which the recommendations are derived (from crowd-powered recommendations or from the user's collaborators on a shared document); and 2) whether or not the user's collaborators are directly identifiable in individual recommendations. We then elicited participants' reactions towards the problem space and each feature recommendation type. We analyzed participant feedback inductively and saw themes emerge related to the participants' different goals for feature discovery, preferences for seeing recommendation from collaborators, and perceptions of how much time they wanted to

invest in such a system. We used these initial insights to inform our design space, which we discuss in the next section.

### 3.3 Design Space Dimensions

The informal formative study provided new insights on potential benefits and drawbacks of tools that raise feature awareness based on the user's collaborators working on shared documents. We do not provide a comprehensive description of these findings here (in part because of some overlap with the elicitation study findings described later). Instead, in this section we discuss how we used the study findings, the related work, and the authors' own experiences to derive a design space. We describe each dimension and provide relevant participant quotes for those motivated by the informal study.

**D1: Number of active collaborators**: Our informal formative study suggested that some participants were more interested in the features that specific individuals were using rather than the features that the majority of their collaborators were using. For example, some participants noted that they would be more willing to try a feature if they perceived their collaborator to be technical savvy. As formative study participant FP00[2] commented "*If it was someone on my team who I know is really tech-savvy, I saw that they used certain functions more, I might pay a little more attention to that*". In contrast, prior work indicates that including more users allows the main user to discover a broader selection of features [50]. Therefore, this dimension investigates whether including a single collaborator (e.g., the technical expert) or more collaborators would help to raise feature awareness: on one end (Fig. 1-**D1**), we have a *single active collaborator*, on the other end we have *all active collaborators*. By active collaborators, we mean the collaborators who have access to the document and actively edit it.

**D2: Number of documents included**: We based this dimension on our (i.e. this paper's authors) experiences. Specifically, while discussing the **D1** dimension, we realized that we often worked with the same group of collaborators to create multiple similar shared documents that follow similar formatting guidelines. For example, the same group of collaborators could work on multiple presentations. This dimension investigates whether including the collaborators' actions from only the current document or other similar shared documents would help the user become aware of relevant features. On one end (Fig. 1-**D2**), we have *the current document* only, while on the other end, we have *all the documents that are shared across the same collaborators*.

**D3: Specificity of comparison**: This dimension is based on existing work on raising feature awareness that explicitly [53,54,58] or implicitly compares [50, 58] the user's individual feature usage habits with the *user community* as an aggregate. On one end (Fig. 1-**D3**), we have tools that *explicitly compare* the user's actions with their collaborators (e.g., with the use of visualizations). The goal of these tools is to help the user reflect on their actions and adjust their habits. On the other end, we have tools that *implicitly compare* the user's actions with their collaborators' to highlight relevant features.

**D4: User involvement**: An early motivation of this work was to investigate tools that raise feature awareness based on the user's collaborators while completely minimizing the user involvement. However, our formative study suggested that users might be willing to invest more time using these systems under specific circumstances. One example that our participants gave

---

[1] We included figures of the prototype in the supplementary material.

[2] We use FPXX to refer to a participant in our formative study

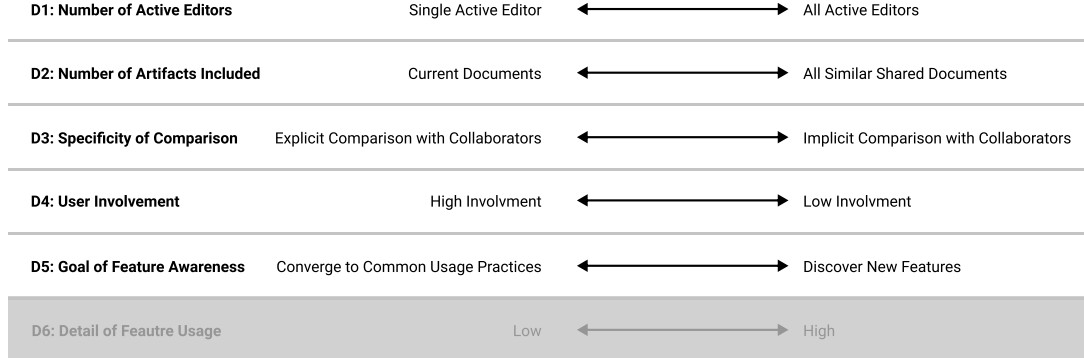

Figure 1: We identified five design dimensions that we used to generate the five design concepts of feature awareness tools: **D1:** Number of Active Collaborators, **D2:** Number of Documents Included, **D3:** User Involvement, **D4:** Specificity of Comparison, and **D5:** Goal of Feature Awareness. Subsequent to the elicitation study, we expanded the design space by adding D6: Detail of Feature Usage.

was for asking follow-up questions regarding the highlighted feature. FP01 said, "*if I have any further questions or a detailed question, I know who I can talk to*". With this dimension we want to investigate the amount of involvement that the user and their collaborators need to invest using the tool for the user to discover relevant features. On one end we have *low user involvement* (Fig. 1-**D4**), where the system focuses on showing the relevant features without offering possibilities for further interaction (as in [54, 56, 58, 59]). On the other end, we have *high user involvement* where the user needs to interact with the system and with their collaborators to find the relevant features (solutions that may fall to this end are [38]).

**D5: Goal of feature discovery**: Perhaps the most unexpected observation from our informal formative study was that participants were interested in how collaborator-based recommendations could help them keep the document formatting consistent across the collaborators. They cared about which commands their collaborators were using *regardless* of whether they already knew the commands or not. For example, FP10 commented that collaborator-based recommendations would be useful "*for the sake of consistency, because people will often use different methods in collaborative documents that do make them a bit messy*". This observation is an interesting contrast to prior work [50, 59] that has identified "good" feature recommendations to be novel and useful to the user. While this might be true for crowd-based recommendations, we see that collaborator-based feature recommendations might be perceived as "good" regardless of whether the user is familiar with the recommended feature. This dimensions aims to explore the user's goal in using the tool. On one end (Fig. 1-**D5**), we have tools that aim to highlight features that may be known to the user already, in order to help the user converge on *common software usage practices*. On the other end, we have tools that aim to highlight *novel features* (i.e., only the features that the user has never used before) that are relevant to the user.

## 4  DESIGN CONCEPTS

To explore where user preference lies within the design space (Fig. 1), we created five design concepts that differ along the design dimensions. For these design concepts, we took inspiration from existing tools that raise feature awareness which we then redesigned to emphasize collaborator-based feature awareness. For each concept, we created a video prototype to illustrate how it works and to be able to compare the design concepts in a systematic manner without the

influence of potential implementation biases [4]. We used Figma[3] to create the clickable prototype and a video editor Camtasia[4] to record the user interaction and produce the final video.

By creating our own concepts and video prototypes, we were able to push the design dimensions in specific directions, often exploring their extremes in new combinations [73, 76]. These design concepts synthesize a mix of contrasting ideas into a cohesive collection, applying existing and proposed design approaches in this new context. It is important to note that this is not an exhaustive exploration, i.e., we did not cover all the possible combinations that we could derive from the design space. This would not be feasible without overwhelming the participants of our elicitation study. We thus focused on the combinations we thought were interesting to explore. For example, we did not design a concept that emphasized *explicit comparison* with *single active collaborator* because we believed that such combination would not necessarily prompt the user to reflect on their actions.

To explain the concepts in the video prototype, we asked the viewer to imagine working on a shared document with other collaborators. We presented all concepts as add-ons to the Google Drive Suite [5] (Google Documents, Google Sheets, and Google Slides). We did not focus on one application of the suite because we wanted to show users that they could potentially install these add-ons with any collaborative shared editor. Finally, we noted to participants that these concepts might raise some privacy concerns, for which we would discuss some solutions in the final discussion with them). However, to keep the focus on the design dimensions, we did not explore any privacy-preserving solutions (except for *NewsFeat*) in the video prototypes.

### 4.1  NewsFeat

The design concept *NewsFeat* was created to strongly emphasize the *single active collaborator* (**D1**), *high user involvement* (**D4**), and *implicit comparisons* between the user's command usage and their collaborators' command usage (**D3**). Additionally, *NewsFeat* focuses on the current shared document in **D2**. It positions in the middle of the **D5** *Goal of feature discovery* dimension.

*NewsFeat* allows users to identify potential useful features by allowing the user to see what commands each of their collaborators use. We took inspiration from existing social networks like Twitter and Facebook where the user can follow other users to see their activities. The user first has to send a request and if their collaborators

---

[3] https://www.figma.com/
[4] https://www.techsmith.com/store/camtasia
[5] https://drive.google.com/

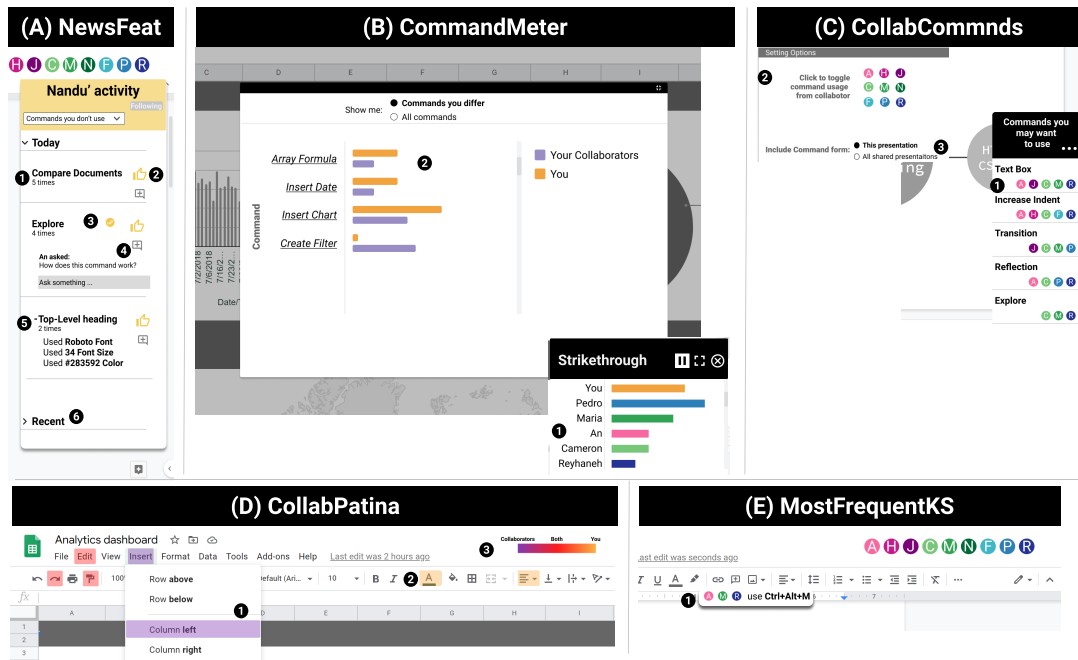

Figure 2: **(A)** *NewsFeat*. **(B)** *CommandMeter*. **(C)** *CollabCommands*. **(D)** *CollabPatina*. **(E)** *MostFrequentKS*.

approve it, the user can see the commands that each collaborator used the same day, including the frequency of use (Fig. 2-**A.1**). By default the user sees the commands that their collaborator used and the user did not (i.e., *implicit comparison* with a *single active collaborator*), although if the user wants they can remove this filter. By allowing users to filter which commands they want to see, *NewsFeat* can be used both to *discover new features* or to *converge to common practices*. In addition, the user and their collaborators can further interact with the system to identify relevant features or to gain more information about one of the commands (i.e., allow *high user involvement*). For example, they can "like" (Fig. 2-**A.2**) a command or ask follow-up questions using the comment button (Fig. 2-**A.4**). The collaborators can also recommend commands that they think are useful. In this case, the recommended command appears with a checkmark next to the command's name (Fig. 2-**A.3**). The collaborators can group a repeated sequence of commands (Fig. 2-**A.5**). For example, imagine a scenario where the collaborator applies the same style (font family, font style, and color) for all the headings. They can group these three commands and give them a specific name. Finally, the user can also see the commands that their collaborators used recently (Fig. 2-**A.6**) in addition to the commands their collaborators used on the same day. They can choose between the commands that their collaborators used last week, last month, or the last six months (not shown in the figure).

## 4.2 CommandMeter

The design concept *CommandMeter* was created to strongly emphasize the *explicit comparison between the user and their collaborators* (**D3**), *high user involvement* (**D4**), and *all active collaborators* (**D1**). Similar to *NewsFeat*, it focuses on the current shared document in **D2**, and has a slight focus on helping users to converge in common usage habits in **D5**.

With *CommandMeter*, the user can identify useful features by comparing their command usage to that of their collaborators through the use of visualizations. By making this *explicit comparison*, the user can reflect on their own behavior and consider whether they want to change their command usage habits. We were inspired by similar systems like Skil-o-Meter [54] that also used visualizations to compare command usage habits between the user

and other members within the same organization. Differently, *CommandMeter* compares command usage habits between the user and their collaborators on a shared document. *CommandMeter* requires *high involvement* as the user has to switch between two views. One view is a collapsible panel on the bottom right corner (Fig. 2-**B.1**). Every time the user selects a command ('Strikethrough' shown in figure), this panel uses horizontal bars to compare the user's and their collaborators' frequency of usage. The second (larger) view offers similar visualizations for all the available commands. In the second view, the user can see all the commands and how their frequency of command usage differs from the average frequency of *all of their active collaborators* (Fig. 2-**B.2**). Finally, they can choose which collaborators they want to include or exclude from their visualizations (not shown in Fig. 2-**B**).

## 4.3 CollabCommands

The design concept *CollabCommands* was created to support users in *discovering new features* (**D5**) based on *all active collaborators* (**D1**). Contrary to the other design concepts, *CollabCommands* strongly emphasizes the possibility to include *all shared documents (D2). It requires* little involvement *from the user (D4).*

With *CollabCommands*, the user can see recommendations derived from their collaborators' usage habits. Drawing inspiration from CommunityCommands [59], *CollabCommands* uses a collapsible panel (bottom right corner) to recommend commands that the user does not use but their collaborators do. Hence, *CollabCommands* offers a quick way for the user to *identify new features* that they might consider using (i.e., requires only *low involvement*). For each command, the tool shows the avatar of the collaborators that are using this command (Fig. 2-**C.1**). The user can further customize the tool if they want. They can choose which collaborators the tool will consider when it decides which commands may be relevant to the user (Fig. 2-**C.2**). Also, the user can decide to include *all other shared documents* in their recommendations (Fig. 2-**C.3**).

## 4.4 CollabPatina

The design concept *CollabPatina* was created to slightly emphasize *explicit comparisons* (**D3**) while minimizing user involvement (**D4**) and it includes all collaborators (**D1**). It focuses on the current

shared document (**D2**) and puts a slight emphasis on converging to common usage practices (**D5**).

*CollabPatina* overlays the current interface with color coded visual indicators to show the user's and their collaborators' feature usage (Fig. 2-**D**). We drew inspiration from the Patina tool [58], but *CollabPatina* is based on the user's collaborators and allows for some extra customisation. *CollabPatina* overlays both the toolbar and the menu with color highlights, indicating which features (commands and keyboard shortcuts) the user frequently uses (Fig. 2-**D.2**) and which features *all of the collaborators* frequently use (Fig. 2-**D.1**). As such, *CollabPatina* requires *low to no involvement* from the user. The color highlights express a visual comparison, but one that is less explicit than in *CommandMeter*. The user can see a color bar on the top of the screen that shows what each color indicates (Fig. 2-**D.3**). When they click the color bar, a setting menu appears (not shown in Fig. 2-**D**) where the user can select whether they want to see color highlights that show the most frequently used commands or highlights that show the most frequently used keyboard shortcuts, or no color highlights.

### 4.5 MostFrequentKS

The design concept *MostFrequentKS* was created to emphasize the discovery of new features (**D5**) (in this case, new keyboard shortcuts), by *implicitly comparing* (**D3**) *all active collaborators* (**D1**). It aims to minimize the *user involvement* (**D4**) and it focuses on the current shared document (**D2**).

*MostFrequentKS* requires low to no involvement from users. When the user selects a menu or toolbar to choose a command, the tool automatically checks if their collaborators frequently use the corresponding keyboard shortcut and shows a notification in the form of tooltip along with the collaborators' avatars (Fig. 2-**E.1**). If none of their collaborators frequently use the keyboard shortcut, then no notification appears. Clicking the toolbar buttons or the menu items will execute the command as it normally would in any scenario. *MostFrequentKS* draws inspiration from tools that use notifications to inform users about the existing keyboard shortcuts [64].

## 5 ELICITATION INTERVIEW STUDY

We used the video prototypes of the design concepts as probes in a semi-structured interview study with 18 participants. The goal of this study was *not* to find a winner among the design concepts but rather to broaden our understanding of the potential benefits and drawbacks of raising feature awareness based on the user's collaborators' application usage, i.e., to assess our general approach to raising feature awareness. We solicited participants' attitudes, reactions, and perceptions of the design concepts, probing the spots in the design space that each concept highlights. In this way, we explored the design dimensions in a semi-targeted way.

### 5.1 Participants

We used a screening survey (available in supplementary material) to recruit participants who had experience collaborating using shared editors. To ensure a diverse sample, we asked participants to mention how often they used collaborative editors, how often they used these editors to work remotely with others, their profession, and the number of collaborators that they worked with. We advertised the study on a mailing list for advertising research studies and stopped recruiting when we reached a saturation point, as is common in qualitative studies. We ended up with 18 participants[6] (10 women, 8 men) between 18-50 years old (the majority were between 18-37 and one 50). The participants had difverse occupations such as software developers, students, receptionists, graphic designers, lighting artists, social workers, and teachers. All participants reported using shared

---

[6]Initially we recruited 20 participants, but we had to exclude 2 participants due to technical issues.

editors like Google Docs to collaborate with others at least one or two times per week. The number of collaborators reported by participants ranged from 2 to 20, with most regularly collaborating with 2 to 4 people.

### 5.2 Procedure

The procedure we followed was based on prior work using RtD approach that used design concepts to elicit user reactions [4,73]. Each session lasted between 60 to 90 minutes. It consisted of three parts: 1) a brief introductory interview focusing on the participants' experiences with collaboration on shared documents, 2) the elicitation part where the participants would see and discuss each design concept, and 3) a final discussion comparing all of the design concepts. One paper author conducted the interviews remotely using Zoom. We recorded all interviews (both audio and video) for later transcription. The participants received $15 per hour as compensation. Our study was approved by an institutional research ethics board.

During the introductory interview, we asked each participant about their experiences with collaborative editors. We asked them about which collaborative editors they used, how often they collaborate with other users, and typical sizes of their teams.

During the main elicitation part, we showed each of the five video prototypes, one at a time in random order. Before showing each prototype, we emphasized that the design concepts are not tied to a specific application, and they should try to reflect on how they would use it within their software of choice. We also told them that although our video prototypes do not address any privacy issues, they should feel free to express privacy concerns. For each video, first, we made sure that the participant understood the concept, and we encouraged them to ask any questions they may have or to replay to video if they wished. Afterward, we asked the participant about their first impressions and their thoughts on each design concept's different aspects. We focused on the aspects that provided insights into the design space. For example, we asked participants if they would use the filtering functionality of *CollabCommands* and *CommandMeter* to include or exclude any collaborator.

During the final part, we asked each participant about their experience across all concepts. We asked them to sort the five concepts from the most to the least preferred and to explain their rationale for their sort order.

### 5.3 Data Analysis

We used thematic analysis [10] to identify recurring themes and patterns from our sessions. We transcribed all sessions and started analyzing them using inductive analysis. Initially, two of the authors coded five transcripts and discussed their codes, and then one author open coded the rest of the sessions. Next, we grouped the codes, and all the authors discussed possible themes and patterns across the groups. We discussed the possible themes over several iterations, focusing on areas that highlighted the potential benefits and drawbacks of raising a user's feature awareness based on their collaborators' use of an application. We used these themes and the participants' feedback on the individual design concepts to identify the approximate relative variation in participants' preferences across the design dimensions.

### 5.4 Findings

Almost all participants (17/18) reported experiences discovering new features while observing their colleagues. In line with prior work [60], the participants found such interactions desirable but rarely happened. For example, P00 explained, "*It's definitely more difficult to find [a new feature] on your own than to observe. Observing is easier.*" As expected, some participants explicitly reported fewer instances of this interaction with the switch to remote working, for example: "*Because it's work from home, we don't really see each other and I don't get to observe their work (P08)*". This participant

went on to talk about using email and messaging to replace such OTS knowledge sharing, yet wishing for an in-application support: "*We'd usually be texting each other or calling each other to inform each other... So, we have to stick to this particular layout, or these other things we have to keep uniform. Instead of doing that communication outside the platform, I think, within the same platform, if you could see this information, I think it will be more efficient*". As such, the participants felt positively about the idea of raising feature awareness based on their collaborators' software use using in-application tools.

### 5.4.1 Overview of User Preference on Design Concepts and Design Dimensions

At the end of each session we asked participants to rank the design concepts from the most preferred to the least preferred. We aggregated all the first and second rankings by participants to identify which concepts participants preferred the most and which the least (this produced 36 ranking data points). *CollabPatina* was the most preferred (13/36) then *NewsFeat* (10/36), followed by *MostFrequentKS* (6/36) and *CollabCommands* (5/36), with *CommandMeter* a clear last (2/36).

It is interesting to note that *CollabPatina* and *NewsFeat* represent different edges in the design space. *CollabPatina* was popular because of its *low user involvement*. This concept's goal was to provide an easy and quick way for the users to see which commands their collaborators use, and more importantly it also shows where the commands are located within the interface. The participants appreciated this functionality because they did not have to spend time locating the commands, which was not the case for *CollabCommands*, *NewsFeat*, and *CommandMeter*. *NewsFeat* was popular because participants could see sequences of commands that their collaborators were using and ask follow-up questions. In contrast, *CommandMeter*, which is also a design concept that requires *high user involvement* was not so popular. It was ranked last most often because it requires high user involvement in order to compare the users' actions to their collaborators.

It is important to note that although *NewsFeat* was well received, participants did raise some concerns regarding feeling self-conscious and micromanaged, which we discuss in Theme 4. Also, the participants were particularly enthusiastic about the ability to see command groupings, but noted that the utility of this aspect of *NewsFeat* would require high user involvement, i.e., the user and their collaborators would need to take the time to create groups of commands. Participants felt that investing this time would be fine under certain circumstances. For example, P05 commented "*... if I want to help new members out in the company, then I would do this. I would group stuff up and then reply to comments and stuff*".

For the rest of the section we discuss themes that emerged across all the design concepts.

### 5.4.2 Theme 1: Raising Feature Awareness Based on The User's Collaborators Could Help Users Converge on Software Usage practices

Consistent with the insights from our informal formative study (Sect. 3.3-**D5**), participants commented on how these tools could help them and their team converge on common software usage practices when working on shared artifacts using feature awareness tools. The participants commented on the usefulness of the concepts to identify similarities and differences in features that their collaborators use to produce a consistent style. For example, P08 commented on why they thought *CollabCommands* could be useful to them and their colleagues "*when I used to work on PowerPoint, we'd usually be texting or calling each other ... to stick to this particular [PowerPoint presentation] layout. Instead of doing that communication outside the platform, I think, within the same platform if you could see this information, it will be more efficient*". P04 highlighted the efficiency of having an in-situ feature usage history displayed in

*NewsFeat*: "*Instead of me having to go and ask, 'What did you do? How did you do this?' I can actually see it in the activity, and it might save a few emails or some back and forth*".

Participants also commented on how they could use these concepts the other way around (for example, the user could help their collaborators converge on common software usage practices). They described, for example, that if a user notices that their collaborators are not using the appropriate commands in a shared document, it could be useful to alert them about it. For example, P04 discussed how they would use *NewsFeat* to help their colleagues "*... if we're stuck on something, if I get to see that, ... oh, okay, this is where maybe somebody got stuck, or why is this being returned to so many times, is there something that we need to revisit in that document itself?*".

Finally, the participants also commented on how these design concepts could help them converge with their own past feature usage. Such a scenario may occur when a user tries to resume a task after a long time and could find it useful to be aware of features they had used in the past. For example, P00 commented on *CollabPatina* "*Well, because I sometimes do things and I forget how I did them. So I like that I can also see how I did things*".

One potential caveat that a couple participants noted was that exposing the user to other collaborators' usage habits may limit their style and creativity. They were concerned that by seeing what features their collaborators are using, they might feel discouraged to use the features they like to use or experiment less with new features. P07 who is a lighting artist had as initial impression of *CollabCommands* was "*It will change my mind to use more and more whatever other people using. It will try to stop creativity, [...]*" while P04 commented on *CommandMeter* "*you might love this feature and want to use it all the time, but the rest of your team might not, and that can be a little tricky because if you're using it and nobody else is using it, then sometimes that's not helpful either*".

### 5.4.3 Theme 2: Raising Feature Awareness Based on The User's Collaborators Could Help Users be More Efficient With Their Tasks.

Some participants felt that they could use these tools to discover more efficient alternatives to do the same task. By efficient alternatives, we do not mean only keyboard shortcuts but also the sequence of steps that other collaborators take to complete the same task. For example, P09 commented when they saw the *CommandMeter*'s visualizations "*For example, if someone is using a command that all of us aren't, meaning something novel and different, that might help us figure out if we can also use that too, maybe it's a better way of doing a task than the version that we've been doing*".

The participants also spoke about wanting to expose their own usage data to help their collaborators discover more efficient alternatives. For example, P02, a project coordinator working with a team of 6, said about *MostFrequentKS*, "*Maybe I would just use this [MostFrequentKS] as a bit of an encouragement for those who might be on the fence about using keyboard shortcuts that, hey, there's actually a bunch of us are using it and this is ... helping us to be more efficient*".

### 5.4.4 Theme 3: Users Want Fine-Grained Control Over Awareness Data Sources

The majority of the participants (14/18) wanted fine-grain control over which subset of collaborators the tool draws feature usage from. They reported that their collaborators might have different roles, such as active editors, viewers, and reviewers. Further, active editors may be in charge of various tasks, only some of which may be relevant to the user. As a result, they felt that the features that the design concept will choose to highlight may not be sufficiently targeted to be valuable. For example, P09 commented, "*There might be people that are just there for review or editing or just viewing*

*purposes so their data will skew it a lot if you don't have the ability to exclude them*".

When we asked participants about which collaborators the tools should include, their opinions differed. Some participants (4/18) wanted to include collaborators based on their role in the document. For example, P15 wanted to include all active editors in their *NewsFeat*: "*probably the owner of the document, and then the main collaborators, and then anyone who's just kind of viewing it or doesn't actually have any [...] stake in the document, [...] then I wouldn't follow them*". Other participants (5/18) wanted to include collaborators that are doing tasks similar to theirs. For example, P09 said "*It's really helpful to be able to include or exclude certain people because [...] everyone is doing different things or there might be certain people that are just on there but not actively working on the documents. So being able to exclude those people from any sort of analytics is important*".

Some participants (5/18) wanted to include individuals based on their perceived expertise or role in the team/company. For example, P00 commented that they would like to include their collaborators who are knowledgeable with the software by using the *CollabCommands* filtering capabilities "*I would include people I know are good at using the type of software that I'm working on*".

Other participants (4/18) did not want to include or exclude any of their collaborators. One possible reason is that, in their teams, all the collaborators have similar roles. For example, P06, a college student, said about *CollabCommands*: "*....it's not like one collaborator is more useful and would have used more commands than another person, necessarily. So yeah, I don't really see a usefulness to that*".

The participants were also interested in having some control over which documents the tools draws the feature usage from. They found this functionality useful if the other documents they included were similar to the current document. P04 commented about this functionality in *CollabCommands*: "*I do find this valuable, because we do work with a lot of similar documents ... and especially because we're always looking to keep things consistent. So, I think having all shared would really help*". Similarly, P09 said, "*I wouldn't want it to do that by default because different documents, ... are trying to do different things ... the commands that I use in one might not necessarily be the same that I use in the other. But the ability to do that, having that option is fine*".

### 5.4.5 Theme 4: Too Detailed Information About the Collaborators' Actions Could Make Users Feel Micromanaged and Self-Conscious

The participants expressed concerns about the detailed information that some design concepts provide. Indeed our design concepts provide information about who used the feature, how often, and how recently to explain why this feature may be relevant to the user. The designs differ on the level of information detail. For example, *NewsFeat* provides more detailed information showing the exact number of times a named collaborator used a command *on the same day*. On the other hand, *CollabPatina* used color-codes highlights to imply the frequency of use of the user's collaborators without identifying the collaborators.

Although seeing more detailed information can benefit the user, as discussed in the previous themes, this information could also lead to feelings of being micromanaged and could cause anxiety among users. For example, when we prompted P07 about how they feel when they saw their collaborators' avatars, they said, "*when I think about seeing collaborators' names using it, I feel like I am a very picky production manager who's trying to micromanage people and make them work faster*". Similarly, when we asked P00 their reactions regarding the recency of information in *NewsFeat* they said, "*Maybe they can have just a vague recents. [...] I wouldn't prefer an option to share daily because then there's an added pressure*". P14 commented regarding detailed information of frequency: "*If there is*

*a command that I have not been using that often, I would feel that I am not contributing that much*".

Some participants felt that detailed information could affect their decision to use a specific design concept and even suggested design changes. For example, P09 said about *NewsFeat*, "*It would definitely make it less invasive if it was just a listing of [the collaborator's] most used commands without any numbers*". Some participants suggested that they would like the ability to hide information to feel less stressed about the information they share. P06 said, "*When I'm giving my permission, maybe I can hide one thing I don't want to show, or things I don't want to show off. Yes. I am giving you permission, but you can see this part, but I will hide the parts I don't want you to see*".

We observed that individual differences related to professional dynamics and personality could affect how users feel about the level of shared details. Problematic professional dynamics such as the position within the organization's hierarchy and the relationship between the user and their collaborators could amplify micromanagement and self-consciousness issues. For example, P07 commented on their experience with their previous manager "*it is just about who are you working with. [...] I've worked with some kind of a person who had psychological disorders, and the minimum mistake you made here will come to your very harsh way and he will give you some psychological difficulties [...] and that's the reason I wouldn't want to see my name is that there too: the blaming point*". Also, the user's personality could affect how they perceive detailed information. If the user is more prone to stressful situations, they may be less open to see and share detailed software usage information. For example, P00 said, "*My boss is super understanding, but I also struggle with anxiety. [...] So to have this other pressure of... I think people deserve a little bit more leniency and every detail shouldn't be shared with the people they're working with*".

## 6   REFLECTION ON THE DESIGN SPACE

The findings from the elicitation study suggest that designers should consider all five dimensions when designing feature awareness tools based on the user's collaborators; none of the dimensions in our design space were shown to be unimportant. To further probe on the participants' preference for each design dimension, we went through the participants' transcripts to specifically look at comments related to the design dimension. We then positioned the participants' comments for each design dimension within the design space (Fig. 3). For example, P2's comment "I don't think I would really care to know who specifically out of my group uses these features" suggested that P2's preference for **D1: Number of active collaborators** leaned strongly towards *all active editors*. In the rest of this section, we reflect on our key findings on user preference within the design space and propose potential design dimensions to expand the design space (as illustrated in Fig. 3), and finally discuss their implications to drive future system designs.

We saw that most participants do want to include only a subset of data sources that the feature awareness tool draws from; they want the ability to control which collaborators (**D1**) and documents (**D2**) are included / excluded. This is an example where participants did not show preference for either end of the spectrum (Fig. 3 - D1 & D2). As a design implication, we imagine an interface that includes by default all collaborators and the current document, while easily allowing further control with interactive widgets.

We also observed that users had a strong preference for implicit comparison (Fig. 3 - D3) and generally prefer to have as little involvement as possible (**D4**). As a design implication, we propose that a system must make it easy for users to locate highlighted features within the interface. This can be accomplished with a solution like *CollabPatina* or with a hybrid solution that lists the highlighted features like *CollabCommands* but provides additional support for locating the feature when the user interacts with it in the list. Beyond

locating a feature, participants are willing to have some involvement for features they deem to be especially valuable; for example, they will ask follow-up questions or actively recommend features to their collaborators (as in *NewsFeat*). Thus, with respect to the *user involvement* design dimension, participants had a preference for the low involvement end of the spectrum but there were some varying opinions (Fig. 3 - **D4**). As a design implication, the system should maximize the information related to the highlighted feature while minimizing user involvement. However, the system should provide non-trivial information, if the user wishes to interact more with it.

Finally, we observed that participants expressed a strong interest in using these tools to both find new features and to help them and their collaborators adopt common feature usage practices **(D5)**. We see, therefore, that participants saw value in being exposed to the features that their collaborators use, both the ones that the user isn't aware of and the ones that the user already knows of (Fig. 3 - D5).

Our findings highlight a trade-off between the availability to view detailed usage information (which collaborator used a feature, how often they used it, and how recently) versus feeling micromanaged and self-conscious. Indeed a lot of the benefits highlighted in Themes 1, 2, and 3 depend on the user having access to this information. However, that same information can cause users to feel negatively, as discussed in Theme 4. Striking the right balance on how to present this information is an important design challenge.

Based on our results, we propose to expand our design space by adding *Detail of feature usage* information as an emerging design dimension. At one end is *Low* level of detail, where designers could reveal collaborators' usage by using language (or a visual indicator) that describes the behavior, but avoids specific numerical values (for example, using "frequently used a command" vs. "used the command 20 times"). On the other end, we have *High* level of detail where designers could use precise numbers, dates, and names. An example of *Low* level of detail is *CollabPatina* which uses color-coded indicators to indicate commands that the user's collaborators frequently use, while an example of *High* level of detail is *NewsFeat*. This dimension is not independent of the other dimensions. For example, a design concept cannot offer explicit comparison **(D3)** without using detailed information. Also, it cannot offer the ability to control which collaborators the feature awareness system draws on without a *High* level detail of the collaborators' identities.

We observed that participants preferred a low level of detail, especially on the recency and frequency of feature usage. They were more comfortable with a system that provides more detailed identification information about the collaborators (Fig. 3 - D6). As a design implication, we propose a system that avoids numerical values for frequency and recency of command usage and can allow for a high level of detail for which collaborators used a feature. Our concepts displayed the avatars of the individual collaborators but a future direction is to include other identification information such as role of the collaborator within the company or their technical expertise.

## 7  OVERALL DISCUSSION

Current solutions that are based on individual users [26, 38] require users to stop their current tasks to either have brief video chats or watch targeted video tutorials. Complementary to this approach, we aimed to leverage the user's collaborators to facilitate in-situ feature discovery while minimizing their involvement and task interruptions. Participants perceived collaborator-based feature awareness tools to be valuable and effective for discovering and adopting common usage practices, but also noted potential issues with self-consciousness and micromanagement.

We reflect on the value of our approach in terms of providing remote over the shoulder learning and how it relates to remote learning from crowd communities. We then discuss our key findings with respect to the need for user and collaborator control over usage

information that is shared.

### 7.1  Supporting Remote Over The Shoulder Learning

Software users often rely on their collaborators to learn new features by observing them [60, 70]. But with the increase in remote work, especially during the COVID pandemic, such over the shoulder (OTS) learning opportunities are limited. Most participants noted that, unfortunately, current tools were limited in providing any support for in-situ software learning and knowledge-sharing, forcing them to coordinate back and forth with their collaborators using external applications (e.g., emails or text messages). The insights from our work can help designers tackle the challenge of supporting in-situ "remote over-the-shoulder learning", especially among collaborators working on shared documents. Although some recent work [38] has investigated how to support remote OTS using video chat, it seems more targeted at complex problems. In contrast, our work proposes more lightweight in-situ techniques for raising feature awareness among collaborators. A future direction is to design support systems that combine various types of remote OTS learning that vary in the user involvement they require and the complexity of the task at hand.

### 7.2  Feature Awareness Tools Based on Different User Communities

In this work we have investigated an alternative to crowd-based approaches by relying on direct collaborators for raising feature awareness. Participants found that direct collaborators can identify the features that can help them complete their task efficiently. Interestingly, participants also commented on the idea of exposing their data to help their collaborators (Theme 1) discover features that they know to be useful. Previous work has focused on how the user can benefit from having access to the usage habits of various user communities [53, 56, 59]. Our work highlights that with a more local community, users also see specific benefit to contributing their data.

While our work has focused on the user's direct collaborators rather than the crowd, we do not see the different user communities as competitors. Each user community can help the user in different ways to raise feature awareness and can even complement each other. Feature awareness systems based on the user's collaborators may be best for helping users to identify the features needed in their current context, i.e., the document they are currently working on. In contrast, feature awareness systems based on the crowd may be best for helping users to expand their feature vocabulary beyond the set of features that their collaborators are using.

A possible future direction is to explore hybrid solutions that support feature awareness based on different user communities. There are several potential design challenges herein. For example, how can we visually distinguish the various user communities? How can we allow the user to switch between user communities and customize their system easily? How can hybrid systems help tackle the privacy concerns highlighted in our elicitation study?

### 7.3  Supporting *User* Control of Data Sources Used for Raising Feature Awareness

Theme 3 discussed how the participants wanted control over the data sources used to support their feature awareness (i.e., which subset of collaborators are viewed and the ability to include similar documents in the comparison), for purposes such as tracking a collaborator who has worked on a particular element of document or is technically savvy. One participant commented that determining the collaborators of interest could be a potential challenge. Although we suspect that this is not going to be a problem for an document that the user is actively working on, perhaps it could be a problem when they include similar documents or newly start working on a document that their collaborators have already been working on. In these cases, it could be useful for the system to highlight collaborators of interest (i.e., the collaborators who worked on the same graphical elements,

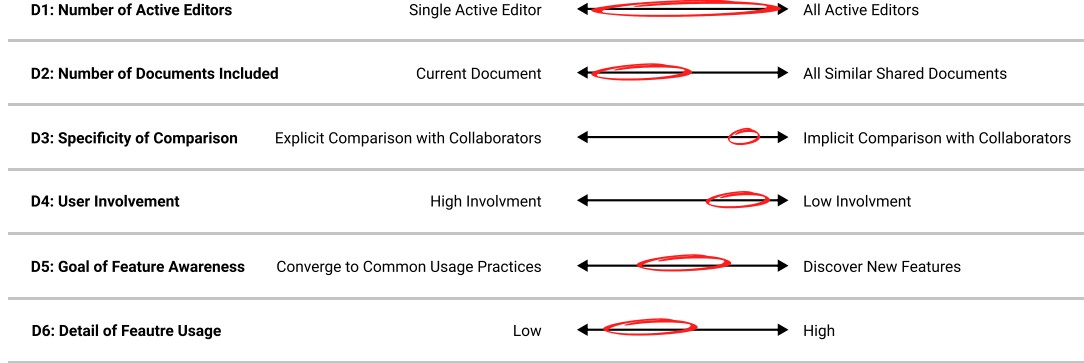

Figure 3: Based on a targeted analysis of the transcripts, we provide a visual representation of the approximate *relative* variation in participants' preferences across the design dimensions. The width of the ellipses provides an indicator of the divergence of opinion.

or the collaborators who are the most active). One potential issue, however, is exposing each collaborator's role can lead to the same problems discussed in Theme 4.

### 7.4 Allowing *Collaborator* Control Over What Information They Share

Theme 4 highlighted how sharing detailed personal feature usage information might make users feel self-conscious, stressed, and micromanaged. However, we also noticed a divergence of opinion, meaning that some participants were more comfortable sharing detailed personal feature usage than others. This divergence could be explained by prior work on factors that affect the users' decision to share personal information to benefit from the system they use.

For example, privacy calculus theory [15] views these decisions as a rational process where users perform a subjective cost-benefit analysis regarding disclosing personal information. This disclosure happens if they anticipate that the benefits outweigh the risks of privacy loss. Work-related to privacy calculus has highlighted some interesting insights such as readiness to embrace new technology [46], self-efficacy [6], trust [52], and amount of involvement [67] that can affect the user's decision to disclose personal information. Furthermore, prior work was identified different personas [46] based on the value users put on the perceived benefits and privacy risks. A future direction is to investigate how these insights apply to our context and the potential design implications.

We also want to explore ways to give the users control over what information they share and the detail of this information. In *Theme 3*, we discussed, for example, a participant who asked for the ability to hide certain commands they used, and we discussed some participants who asked for varying levels of detail sharing. An important future direction is to explore how users can customize the level of detail they share in balancing privacy with the benefits gained by sharing. The challenge is accomplishing this customization in a lightweight manner, given that users generally do not want high user involvement.

One possibility is to give users fine control over *when* they share their feature usage. For example, users can choose to share specific actions by enabling an option in the menu, and when they are done with their task, they can disable the sharing. An alternative is to let users review the highlighted features that the tool has chosen when the user closes the collaborative editor. This solution could help create " learning events" and highlight the features that the user thinks their collaborators would benefit from.

### 8 LIMITATIONS AND FUTURE WORK

The video prototypes used in our elicitation study did not discuss differences in collaborators' roles (e.g., within one organization) because we wanted participants to ground their feedback to their own experiences. However, most participants did not feel that the different roles of their collaborators impacted their perceptions on our design concepts. Only a few participants mentioned certain professional dynamics that may increase the fear of micromanaging, for example if there is a competitive culture in their team. Future work could broaden the participant sample to further probe on other social factors, for example including more diverse age groups and participants with different remote-working experiences, as well as systematically explore how the role of the user within the company (i.e. manager, subordinate) can affect the user's perception of feature awareness based on the user's collaborators.

We designed each of the five concepts as an independent support mechanism, but many of their properties could work in combinations. Combining properties would be an interesting future direction given that two of the most well-received designs *CollabPatina* and *NewsFeat*, offer different functionalities. Our elicitation study used video prototypes to probe participants' reactions and perceptions while reducing biases due to potential implementation issues. One potential direction is to focus on building a feature awareness tool that incorporates aspects from the design concepts that were well received. With this tool, we can conduct longitudinal studies to assess how feature awareness based on the user's actual collaborators will impact the user's actual software usage habit over time.

### 9 CONCLUSION

Our work contributes insights into how we can raise serendipitous feature awareness in remote shared contexts based on a user's collaborators. Drawing upon our informal formative study, prior work, and our own experiences, we created a design space, and then generated five design concepts that exercise this design space of serendipitous feature discovery. Through our elicitation study, we uncovered attitudes and perceptions towards feature awareness tools based on the user's collaborators, highlighting promising design directions and design elements, but also revealing sensitivities that need to be accommodated through careful design. Our work opens up possibilities for new tools that can leverage the user's collaborators' feature usage to provide over-the-shoulder learning in remote contexts. Altogether, it offers a promising direction for addressing feature learnability through improved feature discoverability, a longstanding challenge in HCI.

### 10 ACKNOWLEDGEMENT

This work was supported by the Natural Sciences and Engineering Research Council of Canada (NSERC) "Making it personal: tools and techniques for fostering effective user interaction with feature-rich software" and by European Research Council (ERC) grants n° 695464 "ONE: Unified Principles of Interaction".

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
