# OpenReview forum: "Promoting Feature Awareness by Leveraging Collaborators’ Usage Habits in Collaborative Editors"
_graphicsinterface.org/Graphics_Interface/2022/Conference — GI 2022_

### Official Review · Reviewer_cqV2 · 2022-01-06
**This paper presents the results of an elicitation study to investigate participant reactions to design probes promoting feature awareness for collaborators in “over the shoulder” contexts. Overall, the paper touches on some interesting points but in its current state, is not ready for publication due to inconsistencies in the overall narrative and flaws in the methodology and study design.**

**Rating:** 4
**Confidence:** 4

**Review:**

QUALITY: The quality of the paper is average. In its current state, there are some inconsistencies in narrative throughout the sections, and some major flaws in the methodology and study design. I would consider these major revisions, and therefore recommend a “reject” decision.

ORIGINALITY: The authors motivate the study by stating that there are few “social solutions” to learning about new software features in collaborative work, contrasting this with previous technological solutions that have been ineffective. This was well-motivated and interesting. However, throughout the paper, the study design seems to only take a superficial delve into so-called “social solutions”, without a deep consideration of the social factors that may actually influence realistic adoption of feature awareness tools.

Specifically, the work aims to support “close-knit groups” who are working remotely and “over the shoulder” feature awareness. This seems to assume closeness in collaborator relationships, though this is not explicitly stated or considered in the study design. Specifically, when recruiting participants for the interview study, the study design did not situate participants to the various ROLES they might take on within a team, and how those roles may affect realistic concerns about feature awareness technologies. E.g. whether someone is a close team member that one trusts, versus a supervisor, etc.

Importantly, the authors state that PRIVACY is a dimension that is not considered or explored in the video prototypes. However, this came up many times in the interview findings, where people were concerned about being micromanaged and self-conscious. This isn’t surprising - Given the essential need to consider privacy when developing awareness tools (as already evident in related literature - see below), it is unclear why the authors chose NOT to include privacy concerns in the five dimensions and design probes. In other words, isn't privacy in relation to the roles of team members have, some key considerations that might influence why people do or do not adopt a feature awareness tool like this?

The related work was missing the contextualization of this study within related literature on privacy and awareness systems:
Patil, S., & Kobsa, A. (2009). Privacy considerations in awareness systems: designing with privacy in mind. In Awareness Systems (pp. 187-206). Springer, London.
Patil, S. (2009). Reconciling privacy and awareness in loosely coupled collaboration. University of California, Irvine.
Van den Berg, Bibi, et al. "Privacy in social software." Privacy and identity management for life. Springer, Berlin, Heidelberg, 2011. 33-60.


Finally, I was unclear as to the rigor of methodology which generated the five dimensions. How were participants recruited for the informal formative study?  Was ethics obtained for this study? What were the demographics of participants? Did they have pre-existing relationships with the authors? The 5 design space dimensions also seemed a bit random - some are based on the authors’ own experiences, some are based on related work, and some are based on the informal study quotes. The presentation of this seemed to lack rigor.

SIGNIFICANCE: In its current state, the problem domain this paper looks at (collaborative editors, specifically Google Drive) is not overly convincing as a significant or important problem space. It’s also unclear how generalizable the five dimensions, design probes, or findings are to other remote collaborative domains. To be clear, the authors already clearly state in the title that the paper is about “collaborative editors”. I am just not convinced that 1) this is a significant or worthy domain to explore; and 2) how generalizable these findings are to other remote collaborative domains, like software development, or artistic collaboration, etc. Perhaps it is, but in its current state, this is not well-justified in the text.

CLARITY: The introduction and related work sections are generally well-written. However, there is a lack of a cohesive narrative throughout with regards to a deep consideration of “social solutions” to feature awareness systems (mentioned above). The Findings section is a bit awkwardly written, and the Reflection of the Design Space and Discussion sections have several grammatical errors.

---

### Official Review · Reviewer_9RK1 · 2022-01-13
**This is a well-written, well-structured, and well-presented paper on a sound study that consists of multiple phases with the goal to investigate user reactions to several design concepts for promoting feature awareness in a collaborative environment. I think this paper makes good contributions to the HCI community.**

**Rating:** 9
**Confidence:** 3

**Review:**

This paper focuses on a multi-step study to identify user reactions towards different design concepts for promoting feature awareness in a collaborative environment. The paper is very well-written, well-structured, and well-presented. The related work is informative and seems quite comprehensive. The five design dimensions are clearly described and seem reasonable. The authors’ decisions on the five design concepts using different variations of the design dimensions appear justified. Many findings are interesting with some unexpected observations.  It was a pleasure reading this paper.
A few comments for considerations:
More information about the participants should be provided such as computer skills and education. We should not assume that people use online collaborative tools on a regular basis have equal or strong computer skills.
There’s no description of the video prototypes at all. It would help to have some details of how the prototypes were created. For example, were they paper prototypes, storyboards, PowerPoint prototypes, or software prototypes, etc.
The authors stated that deductive analysis was conducted in addition to the inductive thematic analysis, but there’s no description of how the deductive analysis was performed and what part of the findings was a result of the deductive analysis.

---

### Official Review · Reviewer_Rp2r · 2022-01-15
**Solid work that extends previous work**

**Rating:** 8
**Confidence:** 5

**Review:**

Overall, the work makes a solid contribution, by consolidating previous work that has a strong overlap and collecting common lessons that can be gleaned from this previous work.

Overall, the approach is creative and effective. The study seems to be well conducted and reported. And the discussion is thoughtful and provides useful insights for those working in this area.

I believe this work is a clear accept.

I would make a few comments that might be addressed for a final version.

1) It would be useful to understand which, if any, of the provided findings/themes may have been hinted at in the previous work. It seems likely, although, I am not sure, that at least some of these ideas have previously come up. These could hopefully be added to the discussion.

2) Figure 3, is a provides a nice way to summarize the findings. However, does this simply reflect certain parts of the tools or are there spaces here that none of the tools fall squarely upon? Also, does this describe a new tool that does not yet exist?

3) Current best practices would be no to assume binary gender when reporting participant demographics (e.g., stating "10 participants; 6 female" while short suggests that there are only two genders, which the community is moving away from).

4) The thematic analysis was said to be both "deductive and inductive", but I was confused about how this procedure actually happened. Deductive suggests that there were pre-existing codes? If so, what were those codes, how were they developed and what were they?

Just for potential interest to the authors. This work brought up some findings that reminded me of work in information search by Bateman and Gutwin from about 10 years ago: https://dl.acm.org/doi/10.1145/2441776.2441885

---

### Decision · Program_Chairs · 2022-01-18

Accept